# Recent Molecular Epidemiology of Echovirus 11 Throughout North and West Africa Resulted in the First Identification of a Recombinant Strain from an Acute Flaccid Paralysis Case in West Africa

**DOI:** 10.3390/v16111772

**Published:** 2024-11-13

**Authors:** Ndack Ndiaye, Fatou Diène Thiaw, Adamou Lagare, Thérèse Sinare, Mohamed Lemine Diakité, Serigne Fallou Mbacké Ngom, Ousmane Kébé, Issifi Kollo Abdoulkader, Gassim Cissé, Mohamed Dia, Hermann Nodji Djimadoum, Christelle Ouedraogo Neya, Rakia Boubakar, Issaka Ouedraogo, Landoh Dadja Essoya, Ndongo Dia, Amadou Alpha Sall, Ousmane Faye, Martin Faye

**Affiliations:** 1Virology Department, Institut Pasteur de Dakar, 36 Avenue Pasteur, Dakar 220, Senegal; fatoulayethiaw@gmail.com (F.D.T.); serignefalloumbackengom@esp.sn (S.F.M.N.); ousmane.kebe@pasteur.sn (O.K.); mohamed.dia@pasteur.sn (M.D.); ndongo.dia@pasteur.sn (N.D.); amadou.sall@pasteur.sn (A.A.S.); ousmane.faye@pasteur.sn (O.F.); 2Centre de Recherche Médicale et Sanitaire (CERMES), 634 Bd de la Nation, Niamey YN034, Niger; adamsyn03@gmail.com (A.L.); papakader@yahoo.fr (I.K.A.); boubacar67rakia@gmail.com (R.B.); 3Ministry of Health and Public Hygiene of Burkina Faso, Ouagadougou 7009, Kadiogo, Burkina Faso; theresa.sinare@yahoo.fr (T.S.); christelneya.ouedr@yahoo.fr (C.O.N.); osbareoued@yahoo.com (I.O.); 4Ministry of Health of Mauritania, Avenue Gamel Abdel, Nouakchott 115, Mauritania; brahimdiakitequatre@gmail.com; 5Ministry of Health and Public Hygiene of Guinea, G77P+56P Boulevard de Commerce, Conakry 585, Guinea; cgassim@yahoo.com; 6World Health Organisation Country Office in Mauritania, ILOT K 140-141 Tevragh-Zeina, Route de la Corniche Ouest, Nouakchott 320, Mauritania; djimadoumnodji@gmail.com; 7World Health Organisation Country Office in Guinea, G8Q8+JC6, Corniche N, Conakry 817, Guinea; landohd@who.int

**Keywords:** enterovirus, echovirus E11, genomic sequencing, recombination, phylogeography, West Africa

## Abstract

Echovirus 11 has emerged as a major public health concern, causing sepsis in neonates in many European countries in recent years. In Africa, especially West Africa, where resources and diagnostic capacities are limited, only sporadic cases have been reported. To better understand the recent molecular epidemiology of E11 in West Africa, we characterized twenty-three echovirus 11 strains isolated through the acute flaccid paralysis and environmental surveillance systems for polio from 2013 to 2023, using high-throughput sequencing. Our data are noteworthy due to identifying for the first time a recombinant strain from an acute flaccid paralysis case and represent the first focus to date on molecular characterization of echovirus 11 in West Africa. Moreover, our data show that echovirus 11 diverged from 1970 (95% HPD range, 1961–1979) and evolved into four distinct clades, with the virus spread from West Africa to Europe, exhibiting two introductions in France around 2017, from Senegal and Guinea. Furthermore, the in silico analysis reveals four non-conservative amino acid substitutions in the VP1 sequences of the European strains associated with neonatal sepsis in newborns and a conserved amino acid motif in the VP1 protein toward enterovirus genotypes. Our data provide new insights into the epidemiology of echovirus 11 and point to the crucial need to implement specific surveillance programs targeting non-polio enteroviruses for the rapid identification of emerging or re-emerging enterovirus species, particularly in Africa.

## 1. Introduction

Enteroviruses (EVs) are the most common circulating viruses worldwide and their transmission generally occurs through the fecal–oral route but may also occur via respiratory droplets [1]. Other modes of transmission, such as respiratory and materno-fetal routes, have also been reported [2]. They are associated with a wide spectrum of illnesses, ranging from mild non-specific symptoms, febrile illness, and hand, foot and mouth disease (HFMD) to severe neurological CNS infections such as meningitis, encephalitis and acute flaccid paralysis (AFP) [2]. EVs are a large group of positive-sense single-stranded RNA viruses belonging to the *Enterovirus* genus from the family *Picornaviridae* [3]. Small in size, at 15–30 nm, the EV genome has approximately 7500 nucleotides (nt) and is composed of a large open reading frame (ORF) flanked by 5′ and 3′ untranslated regions (UTRs) [3,4]. The ORF can be translated into a 2189-amino-acid-long polyprotein and then cleaved into three polyprotein precursors, P1, P2, and P3. P1 encodes the structural proteins VP4, VP2, VP3, and VP1, while P2 and P3 encode the non-structural proteins 2A, 2B, and 2C, and 3A, 3B, 3C, and 3D, respectively [4]. Sequencing of the VP1 gene is mainly used for EV genotyping as it provides a perfect correlation with the genotype of the strains and an excellent discrimination of all the taxonomic ranks (from species to variants) [5].

Based on the VP1 protein, which is the most diversified region, EVs include fifteen species, of which viruses in four EV species (EV-A to D) and three rhinovirus species (RV-A to C) infect humans [6]. Certain viruses, such as polioviruses, Coxsackievirus A (CVA), Coxsackievirus B (CVB) and echoviruses, have been classified or assigned to EV species [7].

EVs of the B species (EV-B), particularly echoviruses, represent the most common species associated with infections of the central nervous system (CNS) in Asia, Europe, and the USA during the last decade [8,9,10]. Previously reported outbreaks of aseptic meningitis (AM) mainly involved the CVB genotypes and the echovirus types E4, 6, 9, 11, 13, 18 and 30 [11].

Recently, the World Health Organization (WHO) reported an increasing number of severe neonatal infections associated with E11 in many European countries, notably in France, Italy, Croatia, Spain, Sweden, and the United Kingdom [12,13]. These infections were associated with severe neonatal sepsis, hepatic impairment and multi-organ failure in France [14] and severe hepatitis in Italy [15]. E11 is one of the echovirus types most frequently responsible for neonatal infections [16,17,18], and outbreaks due to E11 in newborns are generally associated with high morbidity and mortality [19,20].

To date, there is a scarcity of data on the epidemiology of E11 conducted in Africa, where knowledge and resources are limited. However, previous data from Senegal provided evidence of their circulation in children with AFP [21]. To fill this gap, we investigated the recent molecular epidemiology of E11 in West Africa using genomic sequencing and phylodynamic analyses to better understand the evolutionary mechanisms that have driven E11’s genetic evolution.

## 2. Materials and Methods

### 2.1. Ethical Statement

This study is a component of the Global Polio Eradication Initiative. This study did not involve human participants but rather the use of cell culture isolates of viruses recovered from stool specimens of AFP cases collected through routine poliomyelitis surveillance activities at the instigation of the Word Health Organization (WHO) for public health purposes. All the technical and ethical aspects were approved by the WHO and the Ministries of Health of the countries concerned. The protocol and oral consent were determined as routine surveillance activity by the steering committee of the WHO in compliance with all the applicable national regulations governing the protection of human subjects.

### 2.2. Data Collection

In this study, a dataset including E11’s VP1, capsid and complete genome sequences retrieved from GenBank (https://www.ncbi.nlm.nih.gov/genbank/) (accessed on 21 February 2024) and the Viral Bioinformatics Resource Centre (BV-BRC) (https://www.bv-brc.org/view/Taxonomy/12058#view_tab=genomes) (accessed on 21 February 2024) [21,22], and the newly characterized E11 sequences obtained from NPEV-positive isolates identified from 2013 to 2023 at the Inter-country Reference Laboratory for Polio in Senegal according to the WHO guidelines [23,24], was analyzed.

### 2.3. RNA Extraction and Molecular Testing

The viral RNA was extracted from 200 µL of virus isolates using the QIAmp Viral RNA Mini Kit (QIAGEN, Hilden, Germany) according to the manufacturer’s instructions. The extracted RNA was eluted in a final volume of 60 µL of nuclease-free water. Subsequently, the RNA extracts were confirmed by specific real-time reverse transcriptase polymerase chain reaction (RT-qPCR) using a pan-enterovirus (panEV) RT-PCR assay [25] with the qScript™ XLT One-Step RT-PCR (Quanta Bio, Beverly, MA, USA) according to the manufacturer’s instructions. In addition, differential singleplex RT qPCR assays for the specific detection of echovirus [26] were performed using specific primers and probes with RNA extracts. The experiments were performed using the CFX96 Real-Time PCR system (Bio-Rad, Singapore) and the echovirus-positive RNAs were also stored at -80 °C until further testing.

### 2.4. Complete Genome Sequencing

Echovirus-positive RNAs with threshold cycle (Ct) values lower than 25 were selected and depleted using mammalian ribosomal RNA-specific primers and RNAseH enzyme (NEB). The purified viral RNA was reverse-transcribed into first-stranded cDNA using random hexamer primers with the Invitrogen SuperScript IV Reverse Transcriptase kit (Thermo Fisher, Waltham, MA, USA). Then, after the second strand cDNA synthesis, the fragmented amplicons were tagged using the “indexes” incorporated into the Nextera XT DNA Library Preparation kit (Illumina, San Diego, CA, USA). The purified libraries were quantified using a Qubit 3.0 fluorometer (Invitrogen Inc., Waltham, MA, USA) and normalized to 200 pM before pooling in a single tube. The pooled libraries were loaded on an Illumina iSeq100 instrument using the iSeq100 reagent v2 for 300 cycles.

### 2.5. Sequencing of the Entire Coding Region of the Capsid Protein

For isolates that provided partial sequences or failed using the Illumina sequencing method, amplicons of the entire coding region of the capsid protein were generated using the OneTaq One-Step RT-PCR kit (New England Biolabs, Ipswich, MA, United States of America) with a specific PCR method, as previously described [27]. Briefly, the PCR was carried out in a final volume of 50 μL, containing 10 μL of the RNA product, 25 μL of the 2× OneTaq One-Step Reaction Mix, 9 μL of nuclease-free water, 2 μL of forward primer (10 μM), 2 μL of reverse primer (10 μM) and 1 μL OneTaq One-Step Enzyme Mix. This reaction’s master mix was amplified using the following conditions: 15 min at 48 °C, 1 min at 94 °C, followed by 40 cycles of 15 s at 94 °C, 30 s at 55 °C and 4 min at 68 °C and a final extension step at 68 °C for 5 min. The amplicon size of the PCR products was verified using electrophoresis on a 1% agarose gel containing SafeView^TM^ Classic (Applied Biological Materials, 3671 Viking Way Unit 1, Richmond, BC, Canada) and visualized under UV fluorescence using the Syngene InGenius3 device (Fisher Scientific, Hampton, New Hampshire, États-Unis).

Amplicons of the capsid protein were purified using the AMPure XP magnetic beads and quantified using the dsDNA High Sensitivity Kit on a Qubit 3.0 fluorometer (Thermo Fisher). The purified entire coding regions of the capsid protein were barcoded using the Rapid Barcoding Kit (SQK.110.96) (Oxford Nanopore Technologies, Gosling Building, Edmund Halley Road, Oxford Science Park, UK) with the MRT001 expansion module (Oxford Nanopore Technology) and pooled in a single tube. The libraries were then purified and sequenced on a GridION instrument (Oxford Nanopore Technology, Gosling Building, Edmund Halley Road, Oxford Science Park, UK).

### 2.6. Sequencing Data Analysis

The raw data from both the Illumina and nanopore experiments were processed, merged into a single FastQ file using a bash command-line pipeline, and consensus sequences were generated using the “Genome detective virus tool” software (version 2.40) [28]. The newly characterized sequences were analyzed using the online Basic Local Alignment Search (BLAST) program (https://blast.ncbi.nlm.nih.gov/Blast.cgi (accessed on 29 February 2024) to assess their homology with previously available data. In addition, the genotype was confirmed using the online Enterovirus Genotyping Tool (RIVM), version 0.1. program (enterovirus/typingtool/).

### 2.7. Recombination Analysis

To assess the occurrence of recombination events, similarity plotting and bootscan analysis were conducted on our dataset by using the SimPlot program (version 3.5.1) [29] with a 400 nt window moving in 20 nt steps and the Kimura two-parameter method with a transition–transversion ratio of 2 with 1000 resampling. Two related reference sequences of parental genotypes previously available via GenBank and one known outgroup (irrelevant) sequence were included in our dataset and simultaneously analyzed to assess the possible breakpoints of intergenotypic recombination. These reference sequences have been chosen as they were considered to be both representative and phylogenetically informative.

In addition, the RDP5 (Recombination Detection Program version 5) program [30] was also used to detect the potential recombinants confirmed by at least 6 of the 7 selected methods using the default settings, including RDP, GENECONV, Maxchi, Bootscan, Siscan, Chimaera, 3Seq and LARD. Full genome sequences were used as queries for BLASTn and sequences with the highest homologies and complete genomes were used in the recombination analysis.

### 2.8. Assessment of Selection Pressures

The ratio of nonsynonymous to synonymous amino acid substitutions (dN/dS) is a useful indicator of the strength of natural selection acting on protein-coding genes. This is a widely used method to detect positive selection. The dN/dS statistical test allowed us to distinguish diversifying or positive selection (dN/dS > 1) from negative or purifying selection (dN/dS < 1). For this, a total of 2 alignment partitions were performed corresponding to the entire coding region of the capsid and VP1 protein. As the site model, we used single-likelihood ancestor counting (SLAC), which estimated the difference between the nonsynonymous (dN) and synonymous (dS) rates per codon site at the 0.1 significance level. The fast, unconstrained Bayesian approximation (FUBAR) method, which evaluated the episodic positive selection at each site in the alignment at a posterior probability ≥0.9, was also used [31]. Finally, the mixed effects model of evolution (MEME) was also conducted at a 0.1 significance level for estimating the selective pressure changes among codon sites. All three methods were conducted with the HyPhy package implemented in the Datamonkey web server (http://www.datamonkey.org, accessed 14 March 2024) [32,33]. An episode of positive diversifying selection was considered if it was detected by at least two different methods.

### 2.9. Phylogenetic Analyses

Multiple alignments were performed using our dataset, including the newly characterized sequences and previous sequences of the capsid protein, the VP1 protein, complete E11 genomes and one outgroup retrieved from GenBank (https://www.ncbi.nlm.nih.gov/genbank, accessed on 26 February 2024), with the ClustalW method implemented in the BioEdit software (version 7.2.5) [34]. The maximum likelihood (ML) phylogenetic tree was inferred using IQ-TREE (version 1.6.12), with the GTR+F+G4 model as the best-fitted nucleotide (nt) substitution model for our dataset, and the branch support was calculated using ultra-fast bootstrapping with 1000 replications [35]. The ML tree topology was visualized with FigTree (version 1.4.4) [36] and the nodes were supported by bootstrap values.

The root-to-tip divergence as well as the regression slopes and correlations analyses were performed using the TempEst v1.5.3 [37] software to assess the reliable temporal signal of the E11 VP1 sequences. Evolutionary analysis of E11 was performed using BEAST2 v2.7.6 [38] to estimate the mean substitution rate and the time of the most recent common ancestor (tMRCA). A maximum clade credibility (MCC) tree was constructed by incorporating a 10% burn-in period. The median heights were generated from a 50 million generation with a Bayesian Markov chain Monte Carlo (MCMC) approach for phylogenetic reconstruction sampling every 1000 generations. In addition, a general time-reversible nucleotide substitution model (GTR) with a gamma distribution prior to each relative substitution rate and a strict clock model were used to infer the echovirus E-11 virus evolutionary timescale. The analysis assumed a strict clock model and a coalescent Bayesian Skyline as the tree priors. Convergence was confirmed by achieving an effective sample size (ESS) greater than 200, using Tracer v1.7.2. [39].

### 2.10. In Silico Analysis Towards the Vaccine Strains

The amino acid polymorphisms on the E11’s VP1 were assessed using a total of 33 sequences, including African and European strains detected in AFP and sepsis, respectively. In addition, the conserved structures were assessed using the web-based ConSurf server (http://consurf.tau.ac.il/ (accessed on 18 May 2024) [40] with the 21_279_SEN_2021 AFP Senegalese E11 strain protein sequence as a query and the multiple sequence alignment as the input. The multiple sequence alignment (MSA) was built using MAFFT. The degree of conservation was calculated using a Bayesian method with the most-fitted evolutionary substitution model for our dataset, considering nine distinct grades, with grade 1 being the least and grade 9 being the most conserved.

## 3. Results

### 3.1. Characteristics of West African E11 Strains Analyzed in This Study

Twenty-three E11 sequences from West African isolates were obtained, including eight VP3-VP1 proteins, two VP1 proteins, twelve entire coding regions of the capsid protein and one complete genome sequence. Moreover, 4 out of these 23 sequences were identified from the environmental surveillance (ES), especially in 2023 (Table 1). A rate of 52.2% (*n* =12) of sequences originated from Senegal. The median age of these E11-positive AFP cases was 2 years (ranging from 16 months to 5 years) and the sex ratio was 3.6 (with 21.7% missing data (*n* = 5)). The virus has been isolated in West Africa since 2013, with an increase since 2021. Moreover, it is mainly identified between March and November, which corresponds to the summer and rainy season in Senegal, with a higher rate in September (31.5% (*n* = 6)). In addition, all the E11 from AFP were identified from single infections, while a mixture was identified for ES involving E19 and E12.

### 3.2. Recombination and Selection Pressures Assessment

One complete E11 genome sequence from Senegal (21-279-SEN-2021) was generated, with a length of 7357 nucleotides (nt), that encoded a polyprotein of 2452 amino acids. The complete genome was closely related (99% nt similarity) to an E11 strain previously identified in non-human primates (NHP) [41] from Nigeria in 2016 (accession number MN657230.1).

The SimPlot graph revealed a difference in the nt similarity in the non-structural genomic region between the 21-279-SEN-2021 strain and a closely related sequence, suggesting a possible recombination event. Furthermore, the occurrence of a breakpoint at nt position 4062 in the P2 region was detected using the bootscanning method. In addition, this recombination event was confirmed using RDP5 analysis, which showed that the E11 strain 21-279-SEN-2021 sequence from Senegal recombined with the E11 strain isolated in NHP from Nigeria in 2016 (accession number MN657230.1) (Appendix A). However, further study is needed since the Nigerian minor parent strain has also been described as recombinant.

Phylogenetic analyses of the complete genome sequence (A), P1 (B), P2 (C) and P3 (D) coding regions of the 21-279-SEN-2021 strain confirmed the RDP5 findings, with the 21-279-SEN-2021 strain clustering with the NHP E11 sequence from Nigeria (accession number MN657230.1) (Figure 1).

The entire coding regions of the capsid and the VP1 protein were analyzed separately for the estimation of sites under positive diversifying selection, applying four methods to ensure the consistency of these events along the structural gene sequences. Using this approach, we found several sites under strong negative selection in both the entire coding region of the capsid and VP1 protein using the SLAC and FUBAR methods. However, sites under episodic positive selection were only observed for the entire coding region of the capsid protein with the MEME method (*p* < 0.1), and therefore, this event was not considered to be significant (Table 2).

### 3.3. Phylogenetic and Evolutionary Analysis

All the newly characterized E11 nucleotide sequences aligned, revealing a high identity at the nucleotide level. Moreover, the phylogenetic ML tree inferences based on the entire coding region of the capsid protein (Figure 2A) and VP1 (Figure 2B) and partial VP1 (Figure 2C) sequences confirmed the data from the BLAST and RIVM and revealed that the E11 sequences were closely related to previous strains, mainly from Africa, and had nucleotide homology ranging from 80 to 100%. Interestingly, our data showed a monophyletic sub-regional clustering of E11 genotype in Africa, which is distinct from the strains that were recently associated with sepsis in Europe (Figure 2).

The root-to-tip analysis revealed a positive temporal signal for E11’s VP1 protein sequences (Figure 3A). The maximum clade credibility (MCC) tree confirmed the genotyping results (Figure 3B) and revealed that the evolutionary substitution rate for E11’s VP1 protein sequence was 7.637 × 10^−3^ substitutions/site/year. Moreover, this analysis showed that the newly characterized E11 sequences were distinct from recent strains identified from the European countries and were not native to their isolated country. In fact, the routes of virus introductions into the West African continent vary. However, North African (Libya, Algeria) and European (France) countries seem to have been the key geographical sources for these newly characterized E11 sequences. Furthermore, we have not observed a temporal and spatial correlation between clusters, suggesting an active virus circulation between countries.

In fact, our data showed that the estimated time to the most recent common ancestor (tMRCA) of the E11 strains diverged from 1970 (95% HPD range, 1961–1979). Subsequently, the virus evolved continuously in four distinct clades: the overall newly characterized E11 sequence grouped into clades I–III and the recent European strains belonging to clade IV. Our data exhibited virus dynamics not only throughout North and West Africa but also its spread from West Africa to Europe. They showed two E11 introductions in France in around 2017, from Senegal and Guinea.

Subsequently, clade I, including the newly characterized Senegalese sequences, has experienced two introductions to Senegal; one from Ghana through Niger in 2013 (95% HPD, 2012–2014) and another from France through Guinea in 2019 (95% HPD 2018–2020). Clade II, grouping Senegalese and Nigerian strains, originated from Libya in 2009 (95% HPD, 2006–2012) while clade III, including Mauritanian strains, spread from Algeria in 2011 (95% HPD, 2009–2013). In addition, the virus spread from Senegal in 2005 (95% HPD, 2003–2007) and was successively introduced to Mauritania and France. Clade IV seems to be restricted to Europe (Figure 3).

### 3.4. VP1 Amino Acid Sequence Analysis

In silico analysis of the amino acid sequences of the VP1 protein has shown a polymorphism for European strains recently associated with sepsis in Europe, including four non-conservative substitutions located at amino acid positions 42, 89, 157 and 211 (Figure 4A). These data suggest a probable impact of these amino acid transitions on the viral pathogenesis or biological characteristics. Furthermore, data from the ConSurf prediction exhibited a conserved amino acid motif in the VP1 protein, located at aa positions 162–169 and corresponding to the EF loop (Figure 4B).

## 4. Discussion

Within the human EV-B species, echoviruses are the most commonly isolated NPEV associated with neurological diseases such as AFP [21,22,42], AM and multisystem hemorrhagic disease [42]. Epidemics of CNS infections involving echoviruses have been previously reported in several countries, with high morbidity and mortality [43]. From 1983 to 2003, case fatality rates ranging from 16.7% to 20% were reported in the United States among patients admitted for CNS infections due to echovirus genotypes such as E6, E7, E11, E20, and E30 [16]. In addition, CNS infections as AM outbreaks mainly involved the E4, E6, E9, E11, E13, E18 and E30 genotypes [6,13].

The recent E11 outbreak in multiple European countries between July 2022 and 2023 was associated with severe acute and fulminant hepatitis in newborns, affecting mainly males [13,14,15]. In Africa, E11 was rarely reported, so only a few data derived from the AFP surveillance and ES are currently available [21,22,41]. Therefore, the surveillance system for AFP and ES could largely contribute to a better understanding of the recent molecular epidemiology of E11 in Africa. Herein, we reported on the genetic diversity and phylodynamics of E11 using high-throughput sequencing, revealing the first complete sequence from an AFP in West Africa to date.

Although echovirus is commonly associated with both epidemic and endemic patterns of infection in individuals of all ages, all the E11-positive isolates from our study included children under 6 years (Table 1), indicating the impact of infections in this vulnerable age group. In addition, recent studies have shown E11’s association with severe disease in neonates, particularly during the first two weeks of life and in those born prematurely [13,14,15]. However, severe complications such as fetal infections in utero have also been associated with disease and/or death [44]. Indeed, human neonatal Fc receptor (FcRn), which is the uncoating receptor for major EV-B [45], is highly expressed on syncytiotrophoblasts, the fetal-derived cells that comprise the outermost cellular barrier of the human placenta and that directly contact maternal blood [46]. Moreover, FcRn can mediate severe E11 infection in suckling mice, but only when they are interferon-deficient, suggesting that interferon signaling may be important for age-related susceptibility [47].

Furthermore, the notion that males are more likely to contract NPEV infection, as previously reported [48], was established in our study (Table 1). Nevertheless, there is a crucial need for further study on sex as a risk factor for enterovirus infections.

Even if NPEVs circulate year-round with a peak in summer [49], we have shown that E11 was mainly identified between March and November (Table 1), suggesting that the prevalence of these viruses could be associated with climate factors such as high temperature and high humidity, as reported in a previous study in China [45].

In our study, four strains were identified from environmental surveillance, demonstrating not only its importance to complement the AFP surveillance but also its sensitivity as an early warning system for pathogens of concern such as NPEV [46].

Interestingly, the occurrence of recombination (Figure 1 and Appendix A) of the newly characterized complete genome sequence with a strain isolated from non-human primates in Nigeria [41] at the non-structural region suggests a potential intratypic recombination during E11’s evolution. In fact, recombination has been reported as a major mechanism participating in the evolution of enteroviruses viruses, especially those belonging to the EV-B species [49,50], and could trigger serious public health problems [51]. In addition, previous data on EV species suggested that intertypic or intratypic recombination frequently occurs in the non-structural regions [52,53], suggesting that these genomic regions represent relatively non-stable units.

The assessment of the selection pressures acting on the E11 genome (Table 2) showed episodes of strong purifying selection in the entire coding region of the capsid and VP1 protein, indicating the stability of these regions. In fact, negative selection pressures, defined as the main evolutionary features avoiding the occurrence of new mutations, were mainly described in the VP1 protein of enteroviruses [54]. Although it could be associated with the pressure from the patient’s immune system, genomic surveillance of EV could be further promoted for rapid identification of possible recombinant strains [54].

The phylogenetic analysis (Figure 2) showed that West African E11 strains formed one group and clustered with previous E11 strains from AFP cases in Ghana in 2017 [55] and from non-human primates in Nigeria in 2016. Similar data have been previously reported for a sample collected in the environment in 2012 [41], suggesting a probable zoo-anthroponotic transmission of enteroviruses from humans to non-human primates. Thus, future studies relying on the assessment of risk factors, such as direct or indirect contact with non-human primates, could be further promoted.

The newly characterized strains showed a high nucleotide identity, which indicated that the West African E11 strains emerged a long time before the strains recently identified in the European Union region. Like other RNA viruses, enteroviruses are characterized by evolutionary mechanisms that shape the genomic diversity by the accumulation of mutations, insertions or deletion due to the lack of proofreading activity of the 3D polymerase [56]. This mechanism is crucial for viral adaptability, dissemination, and pathogenesis [57]. Our data underscore the importance of better monitoring E11 by establishing a sustainable genomic surveillance program for enteroviruses in Africa. To date, there is a low number of complete African genome sequences (*n* = 502) available in GenBank (https://www.ncbi.nlm.nih.gov/genbank, (accessed on 21 February 2024)), and 96% are poliovirus stains (*n* = 481). In fact, the monitoring of E11 circulation was mainly limited to the AFP surveillance system and based on only partial VP1 sequences. In addition, an assessment of the pathogenicity of the West African E11 strains in comparison to those recently associated with sepsis in Europe is warranted.

Our Bayesian MCMC analyses (Figure 3) allowed us to compare the mean evolutionary rate of E11 (7.637 × 10^−3^ substitutions/site/year), which was faster than those of other enterovirus genotypes such as EV-A71 [58]. Furthermore, we identified the virus’s introduction and circulation in Senegal several years ago, suggesting a silent E11 circulation until its detection in AFP cases in 2013. Further, based on our dataset, E11 has been introduced into West Africa from North African (Libya, Algeria) and European (France) countries. In fact, E11 has been frequently identified as the causative agent of outbreaks in these countries [59,60]. Therefore, E11’s introductions may be related to its high international human mobility, as previously reported [61].

Furthermore, as also described for NPEV [62], it was found that amino acid changes on E11’s VP1 region may affect its viral virulence. Considering that the VP1 is notably the receptor-binding protein of enteroviruses and the major neutralizing antigen, it is therefore a major viral factor responsible for cell, tissue and host tropism [63]. Thus, mutations occurring in this region might affect the binding and the uncoating process of E11 and increase its transmission ability.

In addition, one amino acid substitution was located in the critical binding regions for neutralizing antibodies that correspond to the BC loop (located at amino acids 57 to 89), which have been associated with virus escape to the immune system [64]. However, their significance and mechanism require further investigations.

The increasing and continuous identification of emerging and/or re-emerging enteroviruses point to the crucial need for vaccine development. The conserved amino acid motifs in the VP1 protein could not only provide more advantages in the development of a broad-spectrum anti-enteroviral drug but also in vaccine design as it plays a pivotal role in the virus’s life cycle.

## 5. Conclusions

E11 has emerged as a recent public health concern across Europe and better understanding of its molecular epidemiology in West Africa is crucial for preparedness, monitoring and control. Our study is noteworthy due to reporting for the first time on the molecular epidemiology of E11 circulating in West Africa and the first identification of a recombinant strain isolated from an AFP case in West Africa. In addition, our data provide new insights into E11’s emergence and circulation dynamic, and useful information for disease control and prevention. Besides AFP and ES surveillance programs, more specific hospital-based surveillance targeting E11, including genomic sequencing, could also be promoted to assess the burden of E11 in Africa. More experimental investigations focusing on the pathogenicity of the identified E11 isolates in West Africa are warranted.

## Figures and Tables

**Figure 1 viruses-16-01772-f001:**
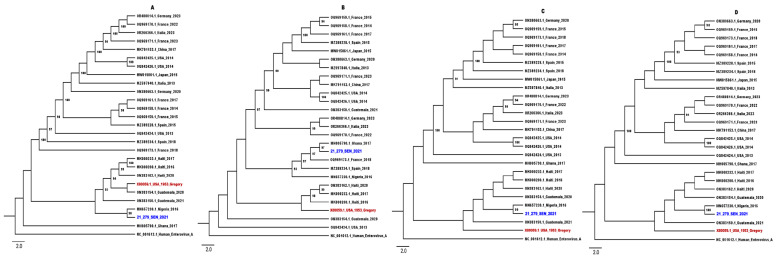
Phylogenetic tree of the complete genome sequence (**A**) (~7381 bp), P1 (~2643 bp) (**B**), P2 (~2000 bp) (**C**) and partial P3 (~2250 bp) (**D**), including 21-279-SEN-2021 and the prototype strain (X80059.1) highlighted in blue and red, respectively. Bootstrap values ≥ 90 are shown on the tree. The scale bar indicates the distances of the branches. The sequences obtained from GenBank included human enterovirus A as an outgroup.

**Figure 2 viruses-16-01772-f002:**
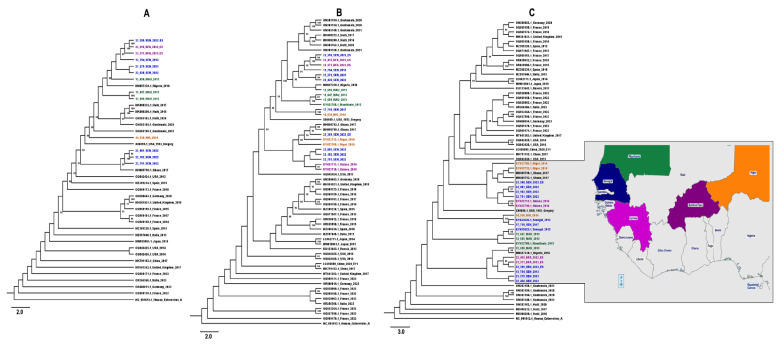
Phylogenetic comparison trees based on (**A**) 41 entire coding region of the capsid protein sequences (~3800 bp), (**B**) 68 VP1 sequences (~780 bp) and (**C**) 70 partial VP1 sequences (~400 bp) from the West African E11 sequences and those obtained from GenBank included human enterovirus A as an outgroup. The characterized sequences are highlighted in blue, purple, green, orange and indigo for Senegal, Guinea, Mauritania, Niger and Burkina Faso, respectively (**C**). The E11 prototype strain (X80059.1) is highlighted in red and bootstrap values ≥ 90 are shown on the tree. The scale bar indicates the distances of the branches.

**Figure 3 viruses-16-01772-f003:**
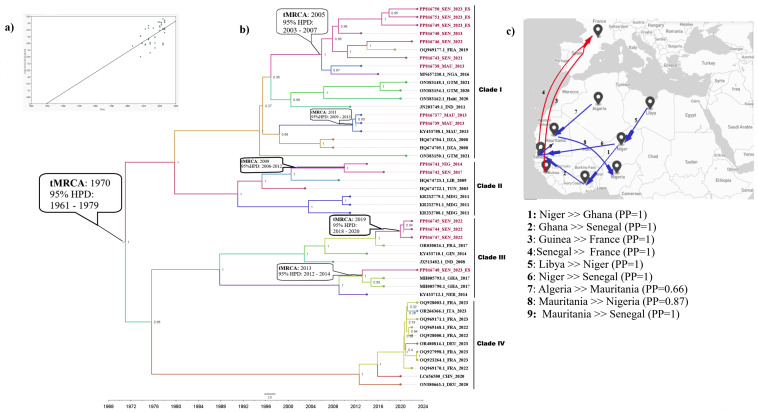
Root-to-tip regression analysis; shown is a linear regression plot for the root-to-tip divergence versus the sampling year (**A**), Bayesian-inferred maximum clade credibility (MCC) phylogenetic tree based on 47 E11 VP1 (**B**) and map showing the spatiotemporal and geographical dispersal of the E11 strains included in the present study (**C**): the grey plots indicate the ocean, blue arrow lines indicate dispersal between West and North Africa, red arrow lines indicate dispersal from West Africa to Europe. The characterized sequences are highlighted in red; the tree branches are colored according to the country of origin. The estimated time to the most recent common ancestor (tMRCA) and posterior probability (PP) are shown in the nodes and the timescale in years is indicated on the *x*-axis.

**Figure 4 viruses-16-01772-f004:**
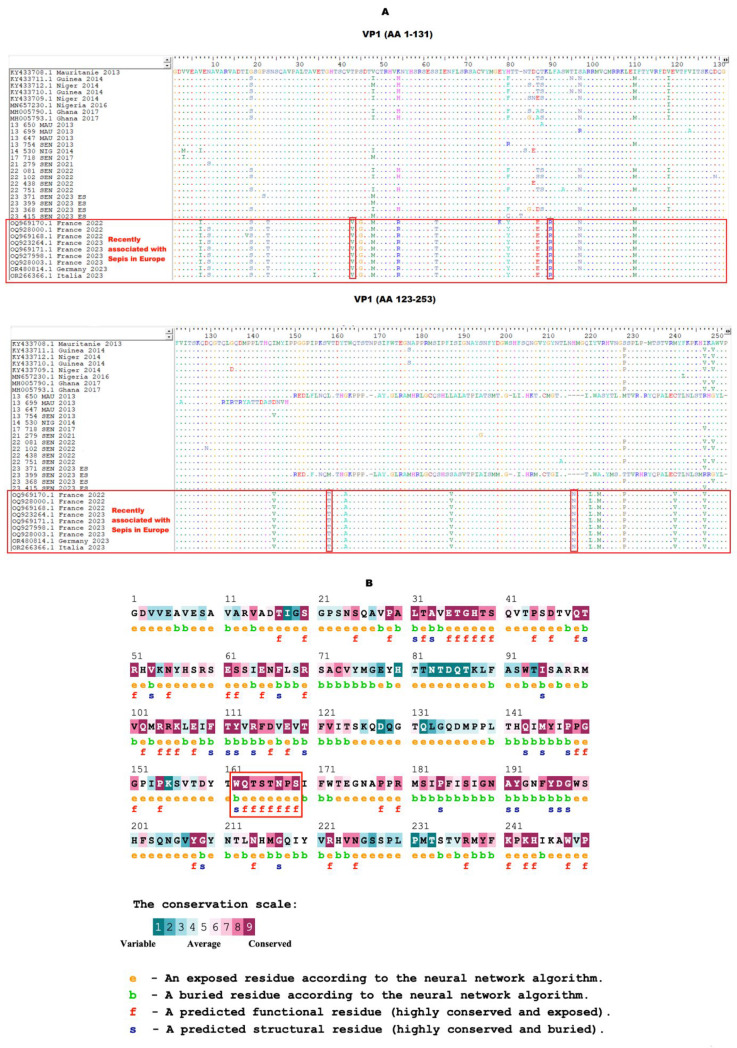
(**A**) In silico analysis of the amino acid sequences of E11’s VP1 protein. The gaps are indicated by a dash (-) and the conserved amino acid residues by a dot (.). Non-conservative amino acid substitutions in strains recently associated with sepsis in Europe are identified by red boxes. (**B**) Evolutionary conservation of amino acids in E11’s VP1 protein using the ConSurf server. The amino acids highlighted in the red box indicate the EF loop’s position.

**Table 1 viruses-16-01772-t001:** Information on the E11 sequences from West Africa analyzed in this study.

Strain ID Number	Collection Year	Country	Case	Age (Month)	Sex	Isolated Month	Sequence Region	Sequence Length (nt)	Accession Number	Reference	Co-Infections
13_650_MAU_2013	2013	Mauritania	AFP	36	M	September	Capsid protein	3804	PP816738	This study	None
13_473_SEN_2013	2013	Senegal	AFP	N/A	N/A	N/A	VP3-VP1 protein	688	KY433623	[23]	None
13_700_MAU_2013	2013	Mauritania	AFP	16	M	September	VP3-VP1 protein	876	KY433708	[23]	None
13_647_MAU_2013	2013	Mauritania	AFP	60	M	August	Capsid protein	3896	PP8167377	This study	None
13_699_MAU_2013	2013	Mauritania	AFP	18	M	September	Capsid protein	3902	PP816739	This study	None
13_754_SEN_2013	2013	Senegal	AFP	24	M	October	Capsid protein	4000	PP816740	This study	None
13_653_SEN_2013	2013	Senegal	AFP	36	M	September	VP3-VP1 protein	690	KY433626	[23]	None
14_530_NIG_2014	2014	Niger	AFP	24	M	September	Capsid protein	3900	PP816741	This study	None
14_240_GUI_2014	2014	Guinea	AFP	24	F	May	VP3-VP1 protein	876	KY433711	[23]	None
14_239_GUI_2014	2014	Guinea	AFP	36	M	May	VP3-VP1 protein	876	KY433710	[23]	None
14_381_NIG_2014	2014	Niger	AFP	19	M	June	VP3-VP1 protein	876	KY433712	[23]	None
14_128_NIG_2014	2014	Niger	AFP	17	F	March	VP3-VP1 protein	876	KY433709	[23]	None
17_787_SEN_2017	2017	Senegal	AFP	N/A	N/A	N/A	VP3-VP1 protein	711	OM827233	[23]	None
17_718_SEN_2017	2017	Senegal	AFP	N/A	N/A	N/A	VP1 protein	783	PP816742	This study	None
21_279_SEN_2021	2022	Senegal	AFP	48	M	March	Complete genome	7357	PP816743	This study	None
22_438_SEN_2022	2022	Senegal	AFP	N/A	N/A	N/A	Capsid protein	3900	PP816746	This study	None
22_081_SEN_2022	2022	Senegal	AFP	24	F	November	Capsid protein	4003	PP816744	This study	None
22_102_SEN_2022	2022	Senegal	AFP	24	M	May	Capsid protein	3800	PP816745	This study	None
22_751_SEN_2022	2022	Senegal	AFP	N/A	N/A	September	Capsid protein	3800	PP816747	This study	None
23_371_BFA_2023	2023	Burkina Faso	ES	N/A	N/A	July	Capsid protein	4000	PP816749	This study	E19
23_399_SEN_2023	2023	Senegal	ES	N/A	N/A	July	Capsid protein	4000	PP816750	This study	E19
23_368_SEN_2023	2023	Senegal	ES	N/A	N/A	July	VP1 protein	804	PP816748	This study	E19
23_415_BFA_2023	2023	Burkina Faso	ES	N/A	N/A	July	Capsid protein	3987	PP816751	This study	E12

F = female; M = male; N/A = not applicable; nt = nucleotide.

**Table 2 viruses-16-01772-t002:** Episodes of positive diversifying selection on the echovirus 11 proteins.

Proteins	Number of Sites Detected by Method				Evidence of Positive Selection
		SLAC (*p* < 0.1)	FUBAR (Posterior Probability ≥ 0.9)	MEME (*p* < 0.1)	
VP1	Sites under negative selection (dN/dS < 1)	134	155	0	NO
	Sites under positive selection (dN/dS > 1)	0	0	0	
Capsid	Sites under negative selection (dN/dS < 1)	877	1150	0	NO
	Sites under positive selection (dN/dS > 1)	0	0	6	

SLAC = Single-likelihood ancestor counting; FUBAR= Fast unconstrained Bayesian approximation. MEME = Mixed effects model of evolution; dN = non-synonymous substitutions; dS = synonymous substitutions; VP1, Capsid = Viral proteins.

## Data Availability

All the data are available in the present manuscript.

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
