# Peer review of "Recent Molecular Epidemiology of Echovirus 11 Throughout North and West Africa Resulted in the First Identification of a Recombinant Strain from an Acute Flaccid Paralysis Case in West Africa"

_viruses, 2024, doi:10.3390/v16111772_

Round 1
Reviewer 1 Report
Comments and Suggestions for Authors
Thanks for the nice paper. It is quite well-written although a balance needs to be found between Intro and Discussion. Sometimes less is more. There are some issues with nomenclature and one questions the images that cannot be seen since they are small. Fig. 3 is slightly overestimate of geographical distribution due to small number of samples, but the presentation is nice and tells the point. Other remarks, which should be easy to correct are below:
Introduction:
-Line 64; this is unclear but you probably mean “viruses in four EV species (EV-A to D) and three rhinovirus species (RV-A to C) infect humans. Note that these viruses when not taxonomical should be written in lower case letters, and no taxonomical unit infects anything, just the viruses.
-Line 66: One might say that certain viruses such as polio- and echoviruses have been classified or assigned to EV species.
-Line 68: missing -s- in species. Use lower case letters with types throughout the text.
-Line 80: check spelling… there is a scarcity of …. on… has been conducted….
-Line 109: Ref 25 is not useful; it is about PhD thesis while the text refers to commercial kit. The method for quantification of E11 should be elaborated.
-Line 118: specify the primers used for making cDNAs.
-Line 268: check spelling for sepsis (Sepis), that is, if you mean “sepsis” - why do you spell it with capital letter? The same in line 310 and Fig. 4.
-Line 323: Echoviruses are echoviruses with lower case letter.
-Line 407: The sentence (chapter) does not read well. Considering…. leads no conclusion.
-Lines 419-439 are out of focus. While important to discuss about Nabs and vaccines, the relevance to the study is missing.
Comments on the Quality of English LanguageIntro and Discussion in imbalance. Some textual corrections needed.
Author Response
Introduction:
-Line 64; this is unclear but you probably mean “viruses in four EV species (EV-A to D) and three rhinovirus species (RV-A to C) infect humans. Note that these viruses when not taxonomical should be written in lower case letters, and no taxonomical unit infects anything, just the viruses.
Response: We thank the author for this comment. This sentence has been rephrased in the revised manuscript.
-Line 66: One might say that certain viruses such as polio- and echoviruses have been classified or assigned to EV species.
Response: This sentence has been rephrased in the revised manuscript.
-Line 68: missing -s- in species. Use lower case letters with types throughout the text.
Response: This remark has been corrected in the revised manuscript
-Line 80: check spelling… there is a scarcity of …. on… has been conducted….
Response: This remark has been corrected in the revised manuscript
-Line 109: Ref 25 is not useful; it is about PhD thesis while the text refers to commercial kit. The method for quantification of E11 should be elaborated.
Response: These remarks have been corrected in the revised manuscript
-Line 118: specify the primers used for making cDNAs.
Response: This remark has been added in the revised manuscript
-Line 268: check spelling for sepsis (Sepis), that is, if you mean “sepsis” - why do you spell it with capital letter? The same in line 310 and Fig. 4.
Response: This remark has been corrected in the revised manuscript
-Line 323: Echoviruses are echoviruses with lower case letter.
Response: This remark has been corrected in the revised manuscript
-Line 407: The sentence (chapter) does not read well. Considering…. leads no conclusion.
Response: This sentence has been rephrased in the revised manuscript for more clarity.
-Lines 419-439 are out of focus. While important to discuss about Nabs and vaccines, the relevance to the study is missing.
Response: These sections have been edited in the revised manuscript according to the reviewer’s suggestion.
Reviewer 2 Report
Comments and Suggestions for Authors
Reviewer Comments
Manuscript number: viruses-3240983
Title: Recent molecular epidemiology of Echovirus 11 throughout North and West Africa exhibited first identification of a recombinant strain
Summary and Recommendation
Enteroviruses are a major cause of morbidity and mortality globally. In this manuscript the authors describe Echovirus 11 genomes detected between 2013 and 2023 as part of the poliovirus surveillance network from cases of acute flaccid paralysis and environmental surveillance. Specifically, they described 23 isolates from Mauritania, Senegal, Niger, Guinea and Burkina-Faso in West Africa. The manuscript is of global interest and deserves to be published as it adds to the body of knowledge. However, the manuscript needs to be revised as some of the claims are contradictory and certain analysis that can alter the current narrative were omitted.
General Comments
1. Line 183: The most abundant data on EV globally in GenBank is partial VP1 using the Nix et al, 2006 assay. Why was this genomic region not explored alongside Capsid and complete VP1 for the analysis. Due to its abundance and ubiquity, for enterovirus molecular epidemiology, partial VP1 always provides better resolution than complete genome, capsid and VP1. Including partial VP1 will help provide a more complete picture of virus dynamics in the region.
2. Please try to increase the resolution of the images.
3. Figure 1: It is common knowledge in the community that recombination is more common in 5’UTR, P2 and P3 genomic regions. If the goal is to explore phylogeny violation as a signal for recombination, standard practice is to explore P1, P2 and P3 independently. Please explain why the authors in figure 1b used P1 and partial P2 instead of only P1. More importantly, please show independent trees for CG, P1, P2 and P3.
4. Figure 2: Why use murine norovirus as an outgroup when there are 14 more enterovirus species from which outgroups can be chosen that will not scramble the alignment. The choice of murine norovirus might have messed up the alignment file and can impact the phylogenetic trees. If possible, can you kindly remake this using an enterovirus that is not a member of species B as outgroup.
5. In lines 361-363 and 377-380 you cite E11 complete genome from west-Africa that precedes this manuscript. Yet in lines 337-338, the conclusion and in the abstract, you claim that this manuscript provides the first complete sequence from west-Africa. That seems contradictory. Please resolve these conflicts.
6. The title also claims this is the first identification of a recombinant strain when the minor parent in figure S1 is also a recombinant strain that was described in West-Africa in 2020 from a 2016 sample.
7. In line 384, you mention ‘high nucleotide identity’ but that data is not shown anywhere in the results. Kindly include the data and make it very visible.
8. Please cite tables and figures in the corresponding places in the discussion section.
9. Line 423: Please consider changing ‘motives’ to ‘motifs’.
10. Please provide more information on the genomes described in figure S1. Kindly make the title more descriptive.
11. Figure S1 shows an E11 isolate that is likely a recombinant and the recombination partner is a virus previously described as a recombinant. Have you considered expanding your search to rule out the possibility that something else might be out there that is more similar to your isolate than the current selected minor parent?
Author Response
General Comments
- Line 183: The most abundant data on EV globally in GenBank is partial VP1 using the Nix et al, 2006 assay. Why was this genomic region not explored alongside Capsid and complete VP1 for the analysis. Due to its abundance and ubiquity, for enterovirus molecular epidemiology, partial VP1 always provides better resolution than complete genome, capsid and VP1. Including partial VP1 will help provide a more complete picture of virus dynamics in the region.
Response: We thank the author for this comment. This remark has been added in the revised manuscript.
- Please try to increase the resolution of the images.
Response: These remarks have been corrected in the revised manuscript
- Figure 1: It is common knowledge in the community that recombination is more common in 5’UTR, P2 and P3 genomic regions. If the goal is to explore phylogeny violation as a signal for recombination, standard practice is to explore P1, P2 and P3 independently. Please explain why the authors in figure 1b used P1 and partial P2 instead of only P1. More importantly, please show independent trees for CG, P1, P2 and P3.
Response: These remarks have been corrected in the revised manuscript
- Figure 2: Why use murine norovirus as an outgroup when there are 14 more enterovirus species from which outgroups can be chosen that will not scramble the alignment. The choice of murine norovirus might have messed up the alignment file and can impact the phylogenetic trees. If possible, can you kindly remake this using an enterovirus that is not a member of species B as outgroup.
Response: This remark has been corrected in the revised manuscript
- In lines 361-363 and 377-380 you cite E11 complete genome from west-Africa that precedes this manuscript. Yet in lines 337-338, the conclusion and in the abstract, you claim that this manuscript provides the first complete sequence from west-Africa. That seems contradictory. Please resolve these conflicts.
Response: This remark has been corrected in the revised manuscript
- The title also claims this is the first identification of a recombinant strain when the minor parent in figure S1 is also a recombinant strain that was described in West-Africa in 2020 from a 2016 sample.
Response: The title has been corrected in the revised manuscript
- In line 384, you mention ‘high nucleotide identity’ but that data is not shown anywhere in the results. Kindly include the data and make it very visible.
Response: This remark has been corrected in the revised manuscript
- Please cite tables and figures in the corresponding places in the discussion section.
Response: These remarks have been added in the revised manuscript
- Line 423: Please consider changing ‘motives’ to ‘motifs’.
Response: This remark has been corrected in the revised manuscript
- Please provide more information on the genomes described in figure S1. Kindly make the title more descriptive.
Response: This remark has been corrected in the revised manuscript
- Figure S1 shows an E11 isolate that is likely a recombinant and the recombination partner is a virus previously described as a recombinant. Have you considered expanding your search to rule out the possibility that something else might be out there that is more similar to your isolate than the current selected minor parent?
Response: We thank the author for this comment. This remark has been added in the revised manuscript.
Round 2
Reviewer 2 Report
Comments and Suggestions for Authors
The authors have improved the manuscript.
Author Response
line 51-54: "They are associated with a wide spectrum of illnesses, ranging
51 from mild non-specific symptoms, febrile illness, Hand Foot and Mouth
Disease (HFMD) to severe neurological CNS infections such as encephalitis and
Acute Flaccid Paralysis (AFP) [2]." EV primarily causes meningitis in terms
of its clinical CNS symptoms, especially in children. The authors missed
meningitis. Ref2 was a virus evolution paper (not a clinical paper) published
11 years ago.
Response: These remarks have been corrected in the revised manuscript
line 64: the authors should clarify the classification is based on sequence
differences in the VP1 gene.
Response: This remark has been added in the revised manuscript.
Line 156-164: similarity plot and bootscan analysis was not described well in
Methods section. Please add the following information: two well-accepted
reference sequences of parental genotypes and one known outgroup (irrelevant)
sequence that were simultaneously analyzed to identify breakpoints of
intergenotypic recombination. The reference sequences should be chosen as
they were considered to be both representative and phylogenetically
informative.
Response: We thank the author for this comment. This remark has been added in the revised manuscript.